# Expected features of the course leader in the rehabilitation healthcare professionals' higher education: A qualitative study on students' perspectives

Benedetto Giardulli[1], Laura Furri[2], Marco Testa[1], Andrea Dell'Isola[3], Gianluca Bertoni[1], Simone Battista[4]*

1 Department of Neurosciences, Rehabilitation, Ophthalmology, Genetics, Maternal and Child Health, University of Genoa, Genova, Italy, 2 School of Medicine and Surgery, University of Verona, Verona, Italy, 3 Department of Clinical Sciences Lund, Lund University, Lund, Sweden, 4 School of Health and Society, Centre for Human Movement and Rehabilitation, University of Salford, Salford, Greater Manchester, United Kingdom

* s.battista@salford.ac.uk

**Data Availability Statement:** All relevant data are within the paper.

## Abstract

Course leaders in rehabilitation healthcare professionals' higher education face challenges stemming from multi-disciplinarity and the co-existence of different stakeholders. So far, the literature mainly attributed to course leaders' managerial tasks, neglecting other fundamental transversal skills. Students represent an essential source of information for understanding the expected characteristics and roles of course leaders in rehabilitation healthcare degree programmes. This study explored students' expected features of the course leaders in the rehabilitation healthcare professionals' higher education. A qualitative interview study was carried out. A group of recent graduates and students of the MSc in 'Healthcare Professionals Rehabilitation Sciences' (University of Verona, Verona, Italy) was recruited using purposive sampling. Data were analysed using 'Reflexive Thematic Analysis' by Braun & Clarke. Ten healthcare professionals agreed to partake in the study (age 30 ± 9; men N = 2; women N = 8). Five themes were generated from the analysis: 1) 'A Collaborative Manager', as students perceived course leaders as non-authoritarian managers who involved all the stakeholders in the decision-making process addressing aspects such as curriculum adjustments, program improvements, and any challenges faced; 2) 'A Diplomatic yet Honest Communicator', as students needed course leaders capable of communicating transparently; 3) 'A Flexible Mediator', as course leaders should actively listen to all stakeholders, mitigating conflicts; 4) 'An Empathic and Available Guide', as students need course leaders available and ready to help; 5) 'An Experienced Healthcare Professional', as students felt course leaders should have a clinical background related to the course they lead. The results of this study suggested that students expect the course leaders to have a wide range of qualities and attitudes about soft (i.e., adaptation, communication, organisation skills, teamwork) and hard skills (i.e., clinical experience, evidence-based practice updated). They expect a course leader to consider all stakeholders' needs and preferences to guarantee course harmony and satisfaction.

**Funding:** The author(s) received no specific funding for this work.

**Competing interests:** The authors have declared that no competing interests exist.

## Introduction

Course Leaders (CLs) are paramount in higher education as they are responsible for students' education and development [1]. They design and oversee degree courses to ensure the highest educational quality [2]. Pursuing academic excellence directly impacts students' satisfaction, a fundamental factor influencing university rankings and institutional appeal [3].

The environment in which the CLs operate within higher education programmes for rehabilitation healthcare professionals, including undergraduate and postgraduate courses, presents significant challenges. It is characterised by multi-disciplinarity and the coexistence of numerous stakeholders, including students, administrative staff, and health and rehabilitation settings. These stakeholders often have differing expectations regarding the management of the courses [4], which are only sometimes clear and shared [5]. Consequently, CLs require distinct skills and characteristics to navigate this complex reality [6].

The literature predominantly emphasises CLs' managerial responsibilities, such as time management and institutional reporting tasks [7]. However, this perspective undervalues and under-recognises CLs' primary role as educators. This limited managerial focus may lead institutions and curricula to overlook other essential transversal skills and characteristics, such as social and communication skills, which can significantly enhance students' overall experience —the ultimate goal of any educational institution [7].

Therefore, it is crucial to consider students' experiences due to their close interaction with CLs, as they can provide valuable insights into the expected characteristics that can enhance the learning experience [8–11]. Qualitative research is fundamental to exploring one's experience [12]. Therefore, the current qualitative study aimed to investigate the expected characteristics of a CL in rehabilitation healthcare professionals' higher education from the perspective of students' experiences.

## Material and methods

### Research design

A qualitative interview study was carried out to explore students' expected features of a course leader in a sample of Italian students or recent graduates at the Master of Science (MSc) degree in 'Rehabilitation Sciences of Healthcare Professions' (University of Verona, Verona, Italy). This MSc is characterised by an interdisciplinary population of healthcare professionals working in rehabilitation (e.g., physiotherapists, speech therapists and occupational therapists). This study followed the 'Declaration of Helsinki' and is reported in line with the Consolidated Criteria for Reporting Qualitative Research (COREQ) [13]. Ethical approval was obtained from the Committee for the Approval of Research on Individuals of the University of Verona (Verona, Italy–date of approval: 13 October 2022, code 15.R1/2022).

### Participants

A group of recent graduates and students of MSc degrees in 'Rehabilitation Sciences of Healthcare Professions' (University of Verona) was recruited through purposive sampling to ensure maximum variation based on the professional background (e.g., physiotherapists and speech therapists), years of experience and specific interest about the topic [14]. Potential participants were identified among those students who chose to conduct a placement in coordination or teaching during the MSc, making them the most suitable to address our research question. The snowball sampling was not adopted. To be included in the study, participants had to be healthcare professionals working in rehabilitation and either current students or recent graduates (within the past year) of the MSc in 'Rehabilitation Sciences of the Health Professions' at

the University of Verona. Finally, participants were provided with a detailed informed consent form encompassing data management practices, privacy protection measures, study-related information, and the overall aim and objectives of the research. Participants were given the written informed consent form prior to participation. Each participant read the form in the presence of the researcher, had the opportunity to ask questions, and if they provided their consent, they could participate in the study.

## Data collection

For deeper exploration, we adopted a semi-structured interview guide (Table 1) created collaboratively by a researcher (SB), a course leader (LF) and a student from the MSc degrees in 'Rehabilitation Sciences of Healthcare Professions' at the University of Verona (NM, see acknowledgements). The interview guide was tested on two students to gain feedback. These students were chosen based on their interest in the topic and were then enrolled for the pilot interviews. Following these pilot interviews, one question was removed from the interview guide as it was repetitive, and others were made more understandable. Once the interview guide was compiled, the interviews were conducted via the Microsoft Teams Platform by a male student of the abovementioned MSc (NM) who was trained by SB in qualitative study and who successfully took the course 'Qualitative Research' at the University of Verona and did not have any close relationships with the interviewees. The interviews were recorded and transcribed automatically by the Microsoft Teams Platform. NM checked the clarity of the transcriptions and saved them in a OneDrive folder at the University of Verona. This folder was accessible only to the MSc student (NM) until the interview transcripts were transcribed and anonymised, as LF and SB are faculty members. Once this process was over, access to this folder was granted to the other research team members. Participants did not review transcriptions for accuracy. Participants were assigned codes based on their interview order, age, gender, and profession to make them anonymous (e.g., P4, 30y, Man, PT). No follow-up interviews were conducted.

## Data analysis

Data were analysed following the six steps of the 'Reflexive Thematic Analysis' (RTA) as reported by Braun & Clarke [15, 16] (Table 2). The choice to use a thematic analysis was driven by the theoretical flexibility and the rich description of data provided by the tool, allowing a

**Table 1. Interview guide.**

| Questions |
| --- |
| 1) Can you tell me, from your perspective, what the strengths and weaknesses of a course leader you have met are? |
| 2) What skills should the course leader have? |
| 3) We now ask you to think of your positive experience with a course leader. |
| 3a) What characteristics did this course leader have? |
| 3b) What were the strengths you found? |
| 4) We now ask you to think of your negative experience with a course leader. |
| 4a) What characteristics did this course leader have? |
| 4b) What were the main difficulties you encountered? |
| 5) What are the attributes that, in general, a course leader should have? |
| 6) What are the attributes that a course leader should instead not have? |
| 7) In general, what do you expect from a course leader? |
| 8) Would you like to add something? |

**Table 2. Steps of the 'Reflexive Thematic Analysis'.**

| Phases | Process | Authors' Involvement | Authors' Actions |
|---|---|---|---|
| 1) Data familiarisation | Two authors immersed themselves in the data to understand the depth and breadth of the content. | BG and SB became familiar with the data in this phase. | Reading and re-reading data set; |
| | | | Listening to the audio recordings; |
| | | | Taking notes; |
| | | | Marking transcripts sections relevant to the research question. |
| 2) Coding | One author started to generate codes to organise the dataset, giving full and equal attention to all data items. | BG re-read the transcripts and developed the recurring ideas into initial codes at the semantic level. Afterwards, all relevant extracts were coded. | Peer debriefing: memos were shared during research meetings for reflexive thoughts; |
| | | | Labelling and organising data items into meaningful groups. |
| 3) Generating initial themes | One author started to generate initial themes by sorting codes and identifying the meaning of and relationships between codes. | BG reviewed the codes and looked for broader patterns of meaning, which were developed into preliminary themes. | Diagramming or mapping to make sense of theme connections; |
| | | | Writing themes and their defining properties. |
| 4) Reviewing and refining themes | Two authors reviewed the initial themes, re-working or eliminating some until finding a set of themes fitting the dataset. | BG and SB discussed and revised the preliminary themes to ensure they represented the codes and the overall patterns identified in the data. | Ensuring there is enough data to support a theme; |
| | | | Re-working and refining codes and themes. |
| 5) Defining and naming themes | Authors developed themes' names and refined them as they could tell a 'story'. | BG and SB refined the definition and name of each theme and wrote a preliminary report outlining each theme. Themes were reviewed to ensure the themes represented participants' experiences and perspectives. | Peer debriefing and team consensus on themes; |
| | | | Cycling the data and the identified themes to organise the story. |
| 6) Producing the report | All authors produced the final report and refined the themes if necessary. | All authors gave feedback and contributed to refining the preliminary analysis. | Writing the final report; |
| | | | Report on reasons for theoretical, methodological, and analytical choices. |

broad meaning of patterns and a rich picture of the participants' experiences. The exploratory theories informing the analysis were an 'experiential qualitative' within a 'realist theoretical' framework as we intended to explore and understand the expected characteristics of the CL to reflect the perception of social reality (rehabilitation healthcare professionals), and to take the fact as voiced in the dataset. In this sense, themes are not in the data waiting but developed by exploring the intersection of the data and the researchers' positioning, skill and interpretative work. The themes were generated by BG, a PhD student in Neurosciences and a physiotherapist, with SB constantly revising the whole process and reflecting upon the themes.

RTA does not follow the (post)positivist paradigm characterised by minimising bias, coding accuracy and the use of different strategies (e.g., data saturation and member checking) to increase data trustworthiness [17]. We used an inductive approach as we did not adopt any predefined framework (i.e. the codebook of the deductive approach). The coding process was conducted on a semantic level of meaning, analysing the explicit or the surface meanings of the data. However, we tried to go beyond these descriptive levels of the data when possible. Different strategies were adopted to ensure study rigour and trustworthiness, such as audit trail of code generation, peer debriefing, and records of all data field notes. During and after each interview, the researcher took field notes–"Memos" and diary—to promote reflexivity, intended as personal reflections relevant to producing knowledge. These memos were shared during research meetings for reflexive thoughts. Moreover, the research team met frequently to refine the themes and subthemes until a consensus on the final themes was achieved. Finally, an audit trail containing meeting notes, analysis discussions, and research decisions was continuously reorganised by the four authors who analysed the interviews to stress the dependability and confirmability of the study [14].

### Research team

BG is a physiotherapist and PhD student in Neurosciences. LF is a physiotherapist and the course leader of the MSc in 'Rehabilitation Sciences of Healthcare Professions'. MT is a physiotherapist, PhD in Rehabilitation Sciences and associate professor. AD is a physiotherapist with a PhD in Musculoskeletal Diseases and associate professor. SB is a physiotherapist with a joint PhD in Neurosciences and Medical Science and a research fellow. BG, MT, AD, and SB identify themselves as men; LF identifies as women. BG and SB are experts in conducting qualitative studies.

## Results

Ten Italian students or recent graduates at the University of Verona agreed to partake in the study (Age (mean and deviation standard): 30 ± 9; 20% Men, N = 2; 80% Women, N = 8; all white Italian). Among the participants, four were physiotherapists (40%), four were speech therapists (40%) and two were psychiatric rehabilitation technicians (20%) (Table 3). From the analysis of the interviews, five themes were developed: 1) 'A Collaborative Manager', 2) 'A Diplomatic yet Honest Communicator', 3) 'A Flexible Mediator', 4) 'An Empathic and Available Guide', 5) 'An Experienced Healthcare Professional'. Quotations and codes that led us to generate the themes are reported in Tables 4–8.

### Theme 1: 'A Collaborative Manager'

By exploring students' experiences, a necessary reported characteristic of CLs was the ability to manage and direct the degree course with a broad vision and a clear educational mission based on shared decision-making, gathering insights from the stakeholders, and assessing alternative resolutions. Hence, we generated the first theme: 'A Collaborative Manager' (Table 4). This theme underscored the importance of participative and non-authoritarian leadership, valorising and involving all relevant stakeholders, especially students and lecturers, in the organisational dynamics of the degree programme. Moreover, students perceived managing a degree programme as demanding and very time-consuming. Therefore, an essential perceived managerial skill of the CL was delegating minor tasks to others and coordinating them to prevent work overload. Finally, as a manager, the CL must maintain clear objectives and a broad vision. In other words, the CL must understand which goals should be prioritised strategically.

**Table 3. Participants' characteristics.**

| Participant | Gender | Age | Health Profession |
|---|---|---|---|
| S1 | Man | 25 | ST |
| S2 | Woman | 27 | PRT |
| S3 | Woman | 25 | ST |
| S4 | Woman | 25 | ST |
| S5 | Man | 41 | PT |
| S6 | Woman | 25 | PT |
| S7 | Woman | 27 | PT |
| S8 | Woman | 28 | PRT |
| S9 | Woman | 44 | PT |
| S10 | Woman | 26 | ST |

P, participant; W, woman; M, man; ST, speech therapist; PRT, psychiatric rehabilitation technician; PT, physiotherapist.

**Table 4. Illustrative data extracts for Theme 1: 'A Collaborative Manager'.**

| Codes defined by researchers | Example of quotes extracted from the interviews |
|---|---|
| **Managing Time** | "The CL must also have managerial skills [. . .] the university indeed has some restrictions in timetable [. . .] However, the CL must also know how to fit them all in the best possible way, so that students and teachers don't waste precious time." (S1, man, 25, ST) |
| **Managing Time** | "The sense of organisation and time management, [. . .] you need to be very good at organising your agenda, keeping up with everything you have to do." (S10, woman, 26, ST) |
| **Having a mission** | "As far as strengths are concerned, certainly tenacity [. . .] to carry out one's mission to the best of one's ability" (S1, man, 25, ST) |
| **Having a mission** | "If there are too many objectives to pursue and there is no priority, there is a risk of [. . .] generating misunderstandings and confusion." (S1, man, 25, ST) |
| **Having a mission/ Collaboration** | "Planning for yourself and others means that you need to have in mind what your and other people's goals are, both in the medium and in the long run, besides knowing how to achieve them." (S1, man, 25, ST) |
| **Determination** | "To get to be a CL, you have to be a very determined person because it's not a job that you can start from one day to another [. . .] you have to be a very determined and resourceful person." (S10, woman, 26, ST) |
| **Collaboration** | "Accepting difficulties and when you are dealing with a problem, you try to solve it together with them [students] without imposing your figure on them." (S9, woman, 44, PT) |
| **Organisational skills** | "They organise and ensure that there is a blueprint. [. . .] They must check that the things the processor says have been correctly anticipated by another professor. In short, they check that all this actually happens." (S3, woman, 25, ST) |
| **Organisational skills** | "The organisation that has the course leader must have a good ability to program, plan all the activities that interface, precisely in that pyramid (degree course)." (S4, woman, 25, ST) |
| **Organisational skills** | "Organisational skills, the ability to delegate [. . .] a good problem-solving ability, that is, finding the right way to solve the problem as efficiently as possible." (S5, man, 41, PT) |
| **Organisational skills** | "Knowing how to plan, organise. . . The entire learning plan for both the student and the study program. [. . .] Conducting an analysis of the needs of the degree program and also an analysis regarding the entire cost-benefit aspect." (S6, woman, 25, PT) |

## Theme 2: 'A Diplomatic yet Honest Communicator'

Every task or facet of the course should be purposefully chosen following a specific rationale. However, to interact effectively with stakeholders and a team, good communication skills are essential to favour the transmission of the correct messages. Hence, we generated the second theme: 'A Diplomatic yet Honest Communicator' (Table 5). The interviewees emphasised the importance of having a CL with good communication skills, the ability to convey messages, and the ability to adapt their language to the audience. Another pivotal factor was that the CLs needed to safeguard their integrity, which meant that CLs were required to be honest and transparent. Moreover, CL needed to recognise their limits and admit when they had no answer to a question.

## Theme 3: 'A Flexible Mediator'

Being able to mediate among different stakeholders, coupled with active listening and receptivity to criticism, was perceived as a crucial characteristic by students, leading us to generate the third theme: 'A Flexible Mediator' (Table 6). Mediating among all the stakeholders around the degree course programme was considered crucial for students, especially concerning conflicts. For instance, the CL could have mitigated uncomfortable situations among students and

**Table 5. Illustrative data extracts for Theme 2: 'A Diplomatic yet Honest Communicator'.**

| Codes defined by researchers | Example of quotes extracted from the interviews |
|---|---|
| Adapting language | "They have to know how to use different languages depending on who they have in front of them." (S4, woman, 25, ST) |
| Adapting language | "Adapting the language, the communication style, depending on the student in front of you, adult, less adult, etc. . . I think that the ability also to somehow adapt to what each person's temperament is, to understand how I can communicate with that person." (S7, woman, 27, PT) |
| Adapting language | "Adapt the language, the communication that is used, depending on the student in front of you, adult, less adult, but always with respect." (S7, woman, 27, PT) |
| Honesty | "Honesty [. . .] In the sense of saying, 'I don't know how to answer your question; I'll look into it and let you know'. I prefer an honest CL rather than one who grasps at straws. I prefer that because it gives me confidence." (S2, woman, 27, PRT) |
| Honesty | "Communicate, communicate, and communicate again. It should not be taken for granted, so be transparent, whether in good or bad, in what is imposed, in what is not imposed, and chosen together." (S7, woman, 27, PT) |
| Honesty | "There must be mutual sincerity because that way you can build a relationship that may last those three years and maybe even more." (S2, woman, 27, PRT) |
| Honesty | "They must be aware of their own limits to work on them. [. . .] By admitting these limits, you can be honest with those in front of you, and, in my opinion, this is very appreciated." (S9, woman, 44, PT) |
| Effective communication | "Effective communication. . . that there is not a kind of monologue from the course leader or something that is dropped from above, but that there is interactivity between the two people." (S6, woman, 25, PT) |
| Credibility | "When you find yourself with people who are fundamentally already in difficulty and ask you for certainties, the fact of continually changing versions on 50 thousand aspects becomes problematic. This had been a source of many conflicts because you lose credibility." (S5, man, 41, PT) |
| Diplomacy | "The course leader must know how to negotiate and be diplomatic. A characteristic that, in my opinion, is very useful." (S8, woman, 28, PRT) |

lecturers. Hence, active listening was a prerequisite to guarantee a suitable mediation between stakeholders, as it allowed CLs to grasp the involved parties' needs and work towards collaborative solutions. Moreover, CLs sometimes had to listen to constructive criticism and try to question themselves. Such flexibility and openness to criticism were perceived as paramount by students.

## Theme 4: 'An Empathic and Available Guide'

Embodying a CL also meant investing time to foster unity among students. A guide capable of empathising with their students was regarded as an exemplary role model, a figure whom students highly appreciated. Hence, we generated the theme 'An Empathic and Available Guide' (Table 7). In their experience, CLs had to bond within the class, offering motivation and helping when needed. In this perspective, CLs were perceived as guides who needed to stay close and present to their students. In addition, CLs, as guides, are also required to spend some time and create activities with the class to create cohesion and mutual knowledge exchange among them. Teamwork and team building were viewed as fundamental activities to which the CL needed to respond actively.

## Theme 5: 'An Experienced Healthcare Professional'

Among the different roles and characteristics gained through students' experiences, CLs must also possess a good reputation and a robust background as healthcare professionals. Students emphasised the crucial need for CLs to maintain a strong clinical background related to the

**Table 6. Illustrative data extracts for Theme 3: 'A Flexible Mediator'.**

| Codes defined by researchers | Example of quotes extracted from the interviews |
|---|---|
| Active Listening | "Here is another weakness: not listening to the other person properly, in the sense that you already start with preconceptions about what a student wants to tell you. [. . .] It has to be an active and participatory way of listening to the others." (S1, man, 25, ST) |
| Active Listening | "Then there is their [CLs'] predisposition to listen to you; this motivates you. [. . .] students need to feel welcomed; they need to be motivated and listened to with empathy." (S2, woman, 27, PRT) |
| Active Listening | "Being open, listening to students, and making them feel comfortable because if the person I'm talking to doesn't put me at ease, I also struggle to express my opinions and give voice to my thoughts." (S6, woman, 25, PT) |
| Active Listening | "They face every difficulty seeing it as an opportunity and know how to listen. Knowing how to listen to students is a useful skill and competence for the course leader." (S3, woman, 25, ST) |
| Active Listening / Reflexivity | "In my opinion, they must have the ability to gather the views of others, namely, students, teachers and external stakeholders. This is to understand the quality of the course and be able to improve it. [. . .] I used to discuss things with the CL, questioning some of the things she was proposing. Simultaneously, she questioned what I was saying, and by doing this, we could reflect upon our proposals growing up together. Because even a CL needs to grow, to understand where they are wrong." (S4, woman, 25, ST) |
| Flexibility | "A flexible coordinator is a coordinator who tries to meet the person in front of them, a person who may have certain needs." (S2, woman, 27, PRT) |
| Mediator | "They must be able to mediate [. . .] they must be able to dismantle difficulties, especially when there are tensions. They must know how to defuse those sparks in the interest of the degree program and the students." (S5, man, 41, PT) |
| Mediator | "Organising the study program effectively and efficiently [. . .] trying to weigh both the needs of the students and the needs coming from above." (S6, woman, 25, PT) |
| Mediator / Flexibility | "Being able to keep all the pieces together [. . .] because the course leader is the one who has to make the effort to bring everything together." (S8, woman, 28, PRT) |
| Reflexivity | "One must have the ability to accept criticism. Therefore, also creating opportunities to receive opinions from everyone." (S9, woman, 44, PT) |

course they lead, meaning they must have a lot of clinical experience and a good reputation among their colleagues, leading to the generation of the theme 'An Experienced Healthcare Professional' (Table 8). This aspect was relevant for students as they considered CLs a helpful guide for their clinical training journey. The interviews also highlighted the need for CLs to update their knowledge and continuously showcase resourcefulness. Students felt that CLs need to be a reliable reference point for discussing the latest evidence available in the literature, demonstrating a proactive approach to staying informed and changing the training clinical practices into the most cutting-edge treatments. However, students also emphasised how being an excellent healthcare professional or lecturer should not be a solid and harsh requirement. The primary expectation was that CLs fulfilled their role, recognising that success in any one of these domains did not necessarily guarantee proficiency in the others.

## Discussions

The current study explored the experiences of a group of Italian healthcare professionals attending an MSc degree to understand the expected characteristics of CLs in rehabilitation healthcare professionals' higher education. In students' experiences, the CLs need to know how to manage and direct the degree course based on a shared-decision making with different stakeholders ('A Collaborative Manager'), to possess good diplomatic and communication skills ('A Diplomatic yet Honest Communicator'), to listen actively to and mediate different stakeholders ('A Flexible Mediator'), to embody an empathetic and present guide for students

**Table 7. Illustrative data extracts for Theme 4: 'An Empathic and Available Guide'.**

| Codes defined by researchers | Example of quotes extracted from the interviews |
| --- | --- |
| A Guide | "I had difficulties in choosing my thesis. The CL guided me in my choice, [. . .] they helped me to do a thesis that was suitable for me [. . .] as they knew me." (S9, woman, 44, PT) |
| A Guide | "And it becomes a guide, also because in some cases, there are people who have difficulties and therefore need someone who lives at the university and who knows how to help them." (S1, man, 25, ST) |
| A Guide | "The course leader must also be a tutor, so they should have tutoring skills. They should be able to guide or direct your path and your university growth journey." (S1, man, 25, ST) |
| A Guide | "They must be a guide [. . .] At the beginning, a student doesn't even know well where they can go, where they want to go. . . In my opinion, they must be a guide who helps you understand and see all possible areas of work." (S8, woman, 28, PRT) |
| Facilitator | "She arrived with this PowerPoint that really enlightened us because she explained everything, step by step, as it should be. She explained what the criteria should be, warmly advised us to reflect on what we had done in the first year. . . She was a clarifier, bringing a tangible tool, essential points that we did not have." (S4, woman, 25, ST) |
| Facilitator | "They must be a facilitator, especially for working students; they must be able to make their lives easier." (S7, woman, 27, PT) |
| Empathy | "The ability to empathise but, at the same time, to guide them, maintaining the right distance. Here, in my opinion, this should be the most important skill that one must have." (S9, woman, 44, PT) |
| Empathy | "The main competence a coordinator must have is empathy. . . So, putting themselves in the student's shoes, but without getting too overwhelmed." (S9, woman, 44, PT) |
| Empathy | "Being healthcare professionals, we must be empathetic. Empathy, we must be empathetic. That cognitive empathy that welcomes the other but then helps them find the solution." (S4, woman, 25, ST) |
| Empathy | "Understanding, knowing well, as if a course leader really had to know each of their students well to understand what is best for them. . ." (S7, woman, 27, PT) |
| Available | "Closeness is having a coordinator in the classroom across, rather than at specific times, with specific references, such as a phone number, office location, or personal e-mail." (S2, woman, 27, PRT) |
| Available | "Very available. . . I mean, they always find time for you." (S8, woman, 28, PRT) |
| Available | "When a student sends an e-mail or requests a meeting, as much as possible, the course leader must respond promptly or at least respond, even if an immediate meeting cannot be arranged. Days or weeks should not pass before getting a response via e-mail or phone. . . So, not only physical presence but also promptness in responding when needed." (S6, woman, 25, PT) |
| Team Building | "There have been several proposals from some coordinators who have engaged in team-building activities, for example, after the creation of group project work, the creation of recreational activities. For me, team-building is essential. I believe that one of the course leader's goals is to create a good group of students with a common objective." (S3, woman, 25, ST) |
| Team Building | "Making the student feel somehow involved in the decision-making process." (S7, woman, 27, PT) |
| Team Building | "Proposes team-building activities using a space where students can gather in a circle." (S7, woman, 27, PT) |
| Comfort | "The course leader should have the ability to accompany students and make them feel comfortable." (S9, woman, 44, PT) |

('An Empathic and Available Guide'), and finally to have a robust and updated background as healthcare professionals ('An Experienced Healthcare Professional').

Embodying the skills and capabilities of a manager and carefully planning the team's tasks and time efficiently are characteristics already present in the literature. Parkin's assertion that a good CL uses "the logic of goals, sequencing and priorities to map what should happen,

**Table 8. Illustrative data extracts for Theme 5: 'An Experienced Healthcare Professional'.**

| Codes defined by researchers | Example of quotes extracted from the interviews |
|---|---|
| Experienced CL / Clinical Experience | "Skills. . .Well, certainly, in my opinion, it is very important to have a graduated CL in the field of the course. So, they need cognitive and theoretical competence on the subject of the degree". (S3, woman, 25, ST) |
| Experienced CL | "In my opinion, it's important that they have a good wealth of experience. . . For me, it would be strange to think of a course leader without classroom experience." (S8, woman, 28, PRT) |
| Updated / Clinical Experience | "It's important that there is no detachment from the clinic. . . In my opinion, they must be prepared and always updated in clinical matters." (S9, woman, 44, PT) |
| Updated | "They really have to keep up with what is happening outside." (S4, woman, 25, ST) |
| Updated | "Ability to update, because a coordinator, in my opinion, must be updated not only at the level of what happens within the class and the degree program but also at the level of scientific journals and articles." (S1, man, 25, ST) |
| Updated | "They must always keep an eye on innovation, in the sense that there may be more valid scientific studies on a specific topic or course, and they must seize the opportunity and integrate it into the degree program." (S4, woman, 25, ST) |
| Updated | "The course leader is responsible for continuous updating and quality of what is offered." (S7, woman, 27, PT) |
| Clinical Experience | "People with experience, who also have a clinical background and have had clinical experiences." (S5, man, 41, PT) |
| Clinical Experience | "If you are a coordinator of a three-year degree, it goes without saying that you must have the technical (clinical) competence of that specific profession. . ." (S8, woman, 28, PRT) |
| Good Reputation | "They must have a certain type of reputation. . . because the coordinator and a good portion of the faculty are healthcare professionals." (S5, man, 41, PT) |
| Stick to only a role | "Some coordinators in the healthcare field divide their time between the degree program and the clinic. I believe this is not good because it brings a coordinator who is physically not there and who has to split their energy between two different things. The clinic is one thing, and organization and training are another." (S8, woman, 28, PRT) |

when and where, and who should be involved" [18] underscored the importance of avoiding time wastage and adhering to a well-structured agenda. This piece of evidence resonated with the views of our interviewees, who emphasised the significance of CLs possessing clear priorities and well-defined goals. Moreover, delegation was necessary for our interviewees, which was in line with another study [19].

Communication skills represented another relevant characteristic, with students attributing value to CLs capable of effectively conveying messages and adapting their language to their audience. Muteswa's work supported this perspective, emphasising the importance of communication skills [20]. Not less important was the honesty and transparency during communication, including the ability to admit limitations, as our students and other authors highlighted as paramount characteristics [18, 21].

Moving beyond managerial and communicative roles, students perceived the CLs as mediators who foster connections among different stakeholders [18], which can help solve conflicts, as reported elsewhere [21, 22]. Active listening, receptivity to criticism, and ability to put themselves in discussion are fundamental characteristics of a mediator from our students' perspectives. These characteristics resonate with the concepts of "Embodied Leadership" and "Enabling Leadership" [18, 23]. The first is characterised by being non-judgmental, listening actively and embracing uncertainty and reflective practice [23]. The second is characterised by considering ideas and strategies suggested by top-down and bottom-up approaches, looking to work with people in teams to pursue innovation and develop a sense of energy and collective commitment [18]. Hence, CLs need to adhere to both leadership styles.

The role of CLs as empathetic guides, capable of investing their time to foster unity among students, was central from students' perspectives, emphasising the need for someone to rely on to solve problems in the degree course [19]. This attitude encompasses the characteristics of "Emotional Intelligence", defined as the 'form of social intelligence that involves the ability to monitor one's own and others' feelings and emotions, to discriminate among them, and to use this information to guide one's thinking and action' [24]. On the other hand, empathy is another fundamental characteristic of the guide role, as it helps mediate between stakeholders and reduce interpersonal conflicts. In this regard, Olga et al. have highlighted how empathy helps to build neutral relations between parties in the educational environment [22]. Moreover, interviewees also felt that the CLs were responsible for team building and class cohesion, which aligns with the results of another study [19].

Finally, the CLs needed to be experienced and updated healthcare professionals with credible reputations. From students' perspectives, this concept meant having " credible " CLs as they were perceived closer to their professional identity. In this way, the students had an example to shape themselves and could lead the education of their academic course. In line with this, the students appreciated when the CLs were updated on the latest evidence-based practice techniques. Also, having a reputation among colleagues was considered an essential element. A study exploring the competencies for effective leadership in higher education highlighted how 'academic credibility' involved reputation and respect [19]. In the same study, 'experience of being an academic' or 'experience in a university setting' were also reported as essential prerequisites [19]. However, it must be noted that the academic and clinical roles should not overshadow or detract time from the primary duties of the course leader role [19].

The insights gained from this study may represent a valuable resource for faculty and policymakers within higher education, especially in the training and development of CL in rehabilitation healthcare fields. Higher education institutions could design targeted training programmes to develop the skills highlighted in our themes for current and aspiring CLs. Future research could explore if these CL characteristics are common among students across different universities, disciplines, and geographical areas and if CLs perceive these characteristics as necessary.

Some limitations of the study need to be addressed. First, only a few rehabilitation healthcare professionals (i.e., physiotherapists, speech therapists, and psychiatric rehabilitation technicians) were represented in our sample. Consequently, the results may not be applied to all the other rehabilitation healthcare professionals. Second, most of the participants were women. Finally, all participants were taken from the same university (University of Verona), so it is impossible to conclude whether our results might transfer to other universities in different geographical areas. Moreover, most of the participants were white women. This consideration is crucial since meanings attached to education might be influenced by gender, ethnicity and place of living [25]. Nevertheless, this study has significantly contributed by providing rich insights into CLs' multifaceted roles and characteristics in the rehabilitation healthcare professionals' higher education. Moreover, the study was designed and conducted with a course leader and a student to increase its relevance. This exploration addresses a topic that has received limited attention in the existing literature, enhancing our understanding of this crucial aspect of academic leadership.

## Conclusions

The results of this study highlighted that students expect the CLs in the rehabilitation healthcare professionals' higher education to have advanced soft skills (adaptability, communication, organisation, teamwork) and hard skills (i.e., clinical experience, evidence-based practice

updated). A huge emphasis was placed on interpersonal and organisational capabilities, under-scoring the importance of effective mentorship for CL training. Additionally, students expected CLs to consider the different needs of all stakeholders, indicating an approach that ensures course quality and satisfaction. These insights contribute valuable perspectives to the literature on academic leadership in the rehabilitation healthcare professionals' higher education.

## Acknowledgments

The authors would like to thank Nicolò Magistrelli for helping to conduct the interviews.

## Author Contributions

**Conceptualization:** Laura Furri, Marco Testa, Simone Battista.

**Data curation:** Benedetto Giardulli, Gianluca Bertoni, Simone Battista.

**Formal analysis:** Benedetto Giardulli, Gianluca Bertoni, Simone Battista.

**Investigation:** Benedetto Giardulli, Andrea Dell'Isola, Gianluca Bertoni.

**Methodology:** Laura Furri, Andrea Dell'Isola, Simone Battista.

**Supervision:** Benedetto Giardulli, Laura Furri, Marco Testa, Andrea Dell'Isola, Simone Battista.

**Writing – original draft:** Benedetto Giardulli, Laura Furri, Marco Testa, Andrea Dell'Isola, Gianluca Bertoni, Simone Battista.

**Writing – review & editing:** Benedetto Giardulli, Laura Furri, Marco Testa, Andrea Dell'Isola, Gianluca Bertoni, Simone Battista.

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
