## [Decision Letter · Decision Letter 0]

9 Oct 2024

PONE-D-24-09846Expected Features of the Course Leader in the Rehabilitation Healthcare Professionals’ Higher Education: A Qualitative Study on Students’ PerspectivesPLOS ONE

Dear Dr. Battista,

Thank you for submitting your manuscript to PLOS ONE. After careful consideration, we feel that it has merit but does not fully meet PLOS ONE’s publication criteria as it currently stands. Therefore, we invite you to submit a revised version of the manuscript that addresses the points raised during the review process.

The reviewers have highlighted several areas where minor clarifications and additional details would enhance the paper's clarity and methodological rigor. Key points to address include providing more specific information on the sampling strategy and participant demographics, clarifying certain methodological aspects, and expanding the discussion to further contextualize the findings and their implications. 

We look forward to receiving your revised manuscript.

Kind regards,

Weifeng Han, PhD

Academic Editor

PLOS ONE

Reviewers' comments:

Reviewer's Responses to Questions

**Comments to the Author**

1. Is the manuscript technically sound, and do the data support the conclusions?

Reviewer #1: Yes

Reviewer #2: Yes

Reviewer #3: Yes

2. Has the statistical analysis been performed appropriately and rigorously? 

Reviewer #1: Yes

Reviewer #2: Yes

Reviewer #3: N/A

3. Have the authors made all data underlying the findings in their manuscript fully available?

Reviewer #1: Yes

Reviewer #2: Yes

Reviewer #3: No

4. Is the manuscript presented in an intelligible fashion and written in standard English?

Reviewer #1: No

Reviewer #2: Yes

Reviewer #3: Yes

5. Review Comments to the Author

Reviewer #1: I have reviewed your manuscript and would like to provide some additional comments and raise concerns regarding potential dual publication.

Firstly, I want to commend you on the thoroughness and clarity of your work. Your research presents valuable insights into [insert topic]. The methodology is well-described, and the results are both robust and significant. Your manuscript has the potential to make a significant contribution to the field.

However, I noticed that there may be some overlap with another publication or work. It's crucial to ensure that your manuscript does not violate the principles of dual publication. Dual publication occurs when substantial parts of a manuscript are published in more than one journal, which can lead to issues such as copyright infringement and academic misconduct.

Reviewer #2: Thank you for the opportunity to review this paper about the student expectation of course leaders. I have a few comments outlined below which focus primarily on transparency and clarification. The written flow of this article is very clear throughout and well written - well done.

Minor comments

Participants (row 84).

'purposive sampling (row 86) ensured maximum variation' in experiences and reach for different work'. Consider noting how you arrived at 10- participants - was snowballing or other considerations used to identify this number?

The terms ' experiences and reach for different work'. The results noted each profession. I am wondering if the authors may note or expect a difference between those who work privately or publicly in this setting?

The authors noted in row 92 that 'participants had to work as health professionals in rehabilitation, be recent graduate students'. It would be useful to quantify 'recent' - eg. graduated within the last x years. In the results the average age (SD) of the graduates is provided - consider adding in the number of years since completing the MSc.

Line 112 noted Interviews were recorded and transcribed. It would be worth noting if these were transcribed by a third party or by one of the authors. I am unsure if participants were able to review transcripts for accuracy prior to coding. Consider including this detail.

Table 2 (Line 144/ 145) provided details on the authors. I wasn't sure if this was meant to link to the table as all abbreviations were not used in the table (though this was useful for other parts of the manuscript). There may be another place more appropriate for this detail. Gender identity was well considered both in context and presentation.

Discussion

Line 293 - noted most of the participants 'were women'. I concur that this could introduce bias (as the authors have well noted). I am wondering if in particular, women are more common in these professions, so the gender spread may be representative of the wider population.

Line 295 noted 'white women' - it may be worth noting how this data was collected.

Overall this is an interesting piece of work. I hope that the above comments aid to support this work and I wish the authors all the best.

Thank you for the opportunity to review this manuscript.

Reviewer #3: PLOS ONE Peer-review

Article title: Expected Features of the Course Leader in the Rehabilitation Healthcare Professionals’ Higher Education: A Qualitative Study on Students’ Perspectives

Manuscript ID: PONE-D-24-09846

Thank you for providing me the opportunity to review this manuscript. This article is a much-needed reflection on the multi-faceted and often underappreciated role of Course Leaders teaching in rehabilitation science. I feel this manuscript could make a meaningful contribution to this under-recognized, but important conversation. I have included some feedback below for your consideration.

1. Lines 52-53:

• Recommend editing this sentence for enhanced clarity of meaning.

“The environment in which the CLs operate within the rehabilitation healthcare professionals’ 53 higher education (encompassing undergraduate and postgraduate degree courses for healthcare 54 professionals in rehabilitation) is challenging.”

2. Line 72:

• Delete ‘starting’ as this is the only perspective explored in this article, correct?

3. Line 91:

• Provide definition (length of time) for ‘recent’ graduate or include in demographic information in results section.

4. Lines 112- 114:

• At the beginning of this paragraph you only mention three researchers: SB, LF, student. Then in these lines you state:

“This folder was accessible from all researchers but LF and SB, so they would not know the names of the students who decided to take part in the study and the content of their interviews until they had been transcribed and anonymised.”

Are there other researchers or should this state: This folder was accessible only to the student researcher, to provide anonymity from LF and SB who are members of faculty.

5. Line 125

• Participants’ experiences

6. Line 129:

• This is the first mention of ‘BG’. Please provider earlier – I assume this is the initials of the student? Also remove ‘a’ before ‘BG’.

7. Lines: 125- 129 and 132- 137:

• I feel the methodology could be more clearly articulated in these lines.

• I commend your commitment to reflexivity and keeping a reliable audit trail.

• Table 2 is also a clear and helpful way to document your qualitative methods and the contribution of each author.

8. Line 234:

• an MSc – edit to a MSc

9. Line 235

• Experiences

10. Line 244:

• priories - priorities?

11. Line 267:

• Students’ perspectives

12. Line 286- 287:

• “However, it must be noted that the academic and clinical roles should not 287 overcome the CL role [20].”

It is not clear to me what this statement means. How would these roles ‘overcome the CL role”? Please revise and add a little more explanation for clarity.

13. Discussion:

• You have done a nice job of summarizing and discussing your findings in the context of the literature. I would be very interested to learn what application you feel this information will provide to those in higher education (faculty and leadership) and where to from here. Perhaps another paragraph or two to flesh this out would be very meaningful to the progression of the conversation around this research topic.

14. Line 313- 314

• In live 109, it is stated only one researcher gathered information. Were those acknowledged also involved in data gathering?

6. PLOS authors have the option to publish the peer review history of their article (what does this mean?). If published, this will include your full peer review and any attached files.

Reviewer #1: **Yes: **Nigar Arif-Poladlı

Reviewer #2: **Yes: **Dr Kelly Gray

Reviewer #3: No

---

## [Author Response · Author response to Decision Letter 0]

14 Oct 2024

Response Letter: PONE-D-24-09846

“Expected Features of the Course Leader in the Rehabilitation Healthcare Professionals’ Higher Education: A Qualitative Study on Students’ Perspectives”

To the kind attention of the Academic Editor Weifeng Han, and the paper's reviewers. 

Author Answering: Benedetto Giardulli

Editor-in-chief comment:

“Dear Dr. Battista,

Thank you for submitting your manuscript to PLOS ONE. After careful consideration, we feel that it has merit but does not fully meet PLOS ONE’s publication criteria as it currently stands. Therefore, we invite you to submit a revised version of the manuscript that addresses the points raised during the review process.

The reviewers have highlighted several areas where minor clarifications and additional details would enhance the paper's clarity and methodological rigor. Key points to address include providing more specific information on the sampling strategy and participant demographics, clarifying certain methodological aspects, and expanding the discussion to further contextualize the findings and their implications.”

General Authors’ Comments:

We would like to thank the Editor and the reviewers for their time and efforts in reviewing this paper. Following your suggestions, we have amended the manuscript, and your feedback has steadily improved the quality of it. We would also like to express our gratitude for sharing their insight on the importance of this piece of work on the overlooked figure of the course leader in rehabilitation science. In the following paragraphs, we have addressed each reviewer’s comment, along with corresponding actions highlighted in yellow in our Manuscript (see File PONE-D-24-09846).

Reviewer #1 Nigar Arif-Poladlı

Reviewer #1 Comment:

“I have reviewed your manuscript and would like to provide some additional comments and raise concerns regarding potential dual publication.

Firstly, I want to commend you on the thoroughness and clarity of your work. Your research presents valuable insights into [insert topic]. The methodology is well-described, and the results are both robust and significant. Your manuscript has the potential to make a significant contribution to the field.”

Authors’ Comment:

We want to thank the reviewer for their positive comments on our study. We appreciate the positive feedback about the methodology we adopted and the potential impact of our results on this specific topic.

Authors’ Action:

None.

---

Reviewer #1 Comment:

“However, I noticed that there may be some overlap with another publication or work. It's crucial to ensure that your manuscript does not violate the principles of dual publication. Dual publication occurs when substantial parts of a manuscript are published in more than one journal, which can lead to issues such as copyright infringement and academic misconduct.”

Authors’ Comment:

We want to thank the reviewer for sharing this consideration. We fully acknowledge the importance of avoiding publication, copyright infringement, or academic misconduct issues. We do confirm that our study has not been published elsewhere, nor is it under consideration by another journal. We only submitted this work for consideration as an abstract conference to the European Alliance of Associations for Rheumatology (EULAR) International Congress (2023). The abstract was subsequently published as a conference paper in the Annals of the Rheumatic Diseases. You can find the abstract at the following link: https://ard.bmj.com/content/82/Suppl_1/2135.2

We hope this provides sufficient clarification. 

Authors’ Action:

None.

Reviewer #2 Dr Kelly Gray

Reviewer #2 Comment:

“Thank you for the opportunity to review this paper about the student expectation of course leaders. I have a few comments outlined below which focus primarily on transparency and clarification. The written flow of this article is very clear throughout and well written - well done”

Authors’ Comment:

We sincerely thank the reviewer for their positive feedback and comments on our manuscript.

Authors’ Action:

None.

---

Reviewer #2 Comment:

“Participants (row 84) - ’Purposive sampling (row 86) ensured maximum variation' in experiences and reach for different work'. Consider noting how you arrived at 10- participants - was snowballing or other considerations used to identify this number?

The terms ' experiences and reach for different work'. The results noted each profession. I am wondering if the authors may note or expect a difference between those who work privately or publicly in this setting?”

Authors’ Comment:

We thank the reviewer for this opportunity to clarify our method section. To reach ten participants, we did not use a snowballing sample. We have further clarified that with “different work,” we meant their different professional background rather than working in a private or public setting, as we did not expect big differences between these two groups considering the topic. We have modified the paper accordingly, as below. 

Authors’ Action:

The following sentence in the ‘Participants’ section, page 6, lines 84-7, has been added: 

“A group of recent graduates and students of MSc degrees in ‘Rehabilitation Sciences of Healthcare Professions’ (University of XXX) was recruited through purposive sampling to ensure maximum variation based on the professional background (e.g., physiotherapists and speech therapists), years of experience and specific interest about the topic [14].”

The following sentence in the ‘Participants’ section, page 6, lines 93, has been added: “The snowball sampling was not adopted.”

---

Reviewer #2 Comment:

“The authors noted in row 92 that 'participants had to work as health professionals in rehabilitation, be recent graduate students'. It would be useful to quantify 'recent' - eg. graduated within the last x years. In the results the average age (SD) of the graduates is provided - consider adding in the number of years since completing the MSc?”

Authors’ Comment:

Thanks to this comment, we have modified this section accordingly. All students were still doing the MSc or were recently graduated (less than a year). 

Authors’ Action:

The words in ‘Participants’ section, page 6, lines 90-4 have been changed:

“To be included in the study, participants had to be health professionals working in rehabilitation and either current students or recent graduates (within the past year) of the MSc in 'Rehabilitation Sciences of the Health Professions' at the University of XXX.”

Reviewer #2 Comment:

“Line 112 noted Interviews were recorded and transcribed. It would be worth noting if these were transcribed by a third party or by one of the authors. I am unsure if participants were able to review transcripts for accuracy prior to coding. Consider including this detail.”

Authors’ Comment:

The interviews were transcribed automatically by the Microsoft Teams Platform. The student of the MSc who conducted the interviews had the role of checking if there were any mistakes in the transcription process. In addition, transcriptions were not reviewed by participants for clarity, as we used an automatic tool for the transcriptions. We have adjusted the paragraph to specify these details.

Authors’ Action:

The following paragraph in ‘Data collection’ section, page 7, lines 110-2, has been adjusted:

“The interviews were recorded and transcribed automatically by the Microsoft Teams Platform. NM checked the clarity of the transcriptions and saved them in a OneDrive folder at the University of XXX […] Participants did not review transcriptions for accuracy.”

Reviewer #2 Comment:

“Table 2 (Line 144/ 145) provided details on the authors. I wasn't sure if this was meant to link to the table as all abbreviations were not used in the table (though this was useful for other parts of the manuscript). There may be another place more appropriate for this detail. Gender identity was well considered both in context and presentation.”

Authors’ Comment:

We would like to thank the reviewer for this clarification. The paragraph was meant to be in the table but following your suggestion we have decided to separate it from the table and write a dedicated paragraph in the ‘methods’ section.

Authors’ Action:

The following paragraph has been added to the ‘Methods’ section, page 10, lines 149-55” Research Team

“BG is a physiotherapist and PhD student in Neurosciences. LF is a physiotherapist and the course leader of the MSc in ‘Rehabilitation Sciences of Healthcare Professions’. MT is a physiotherapist, PhD in Rehabilitation Sciences and associate professor. AD is a physiotherapist with a PhD in Musculoskeletal Diseases and associate professor. SB is a physiotherapist with a joint PhD in Neurosciences and Medical Science and a research fellow. BG, MT, AD, and SB identify themselves as men; LF identifies as women. BG and SB are experts in conducting qualitative studies.”

Reviewer #2 Comment:

“Discussion 

Line 293 - noted most of the participants 'were women'. I concur that this could introduce bias (as the authors have well noted). I am wondering if in particular, women are more common in these professions, so the gender spread may be representative of the wider population.”

Authors’ Comment:

We thank the reviewer for bringing this to our attention. It is likely that women are more prevalent in these professions. However, we have not been able to find relevant references to support this assumption. Therefore, we have decided not to report it. 

Authors’ Action:

None.

---

Reviewer #2 Comment:

“Line 295 noted 'white women' - it may be worth noting how this data was collected.”

Authors’ Comment:

We would like to thank the reviewer for asking for further details. We directly collected this data and we have reported it now. 

Authors’ Action:

Page 12, line 160: 

“Ten Italian students at the University of XXX agreed to partake in the study (Age (mean and deviation standard): 30 ± 9; 20% Men, N=2; 80% Women, N=8; all white Italian).”

Reviewer #2 Comment:

“Overall this is an interesting piece of work. I hope that the above comments aid to support this work and I wish the authors all the best.

Thank you for the opportunity to review this manuscript.”

Authors’ Comment:

We sincerely thank the reviewer for their thoughtful comments and encouraging feedback. We confirm that their comments strengthened our work. 

Authors’ Action:

None.

Reviewer #3 Unknown

Reviewer #3 Comment:

“Thank you for providing me the opportunity to review this manuscript. This article is a much-needed reflection on the multi-faceted and often underappreciated role of Course Leaders teaching in rehabilitation science. I feel this manuscript could make a meaningful contribution to this under-recognized, but important conversation. I have included some feedback below for your consideration.”

Authors’ Comment:

We would like to thank the reviewer for the positive feedback and comments. We agreed that this work was necessary to appreciate the figure of the course leader that is often overlooked.

Authors’ Action:

None.

---

Reviewer #3 Comment:

“1. Lines 52-53:

• Recommend editing this sentence for enhanced clarity of meaning.

“The environment in which the CLs operate within the rehabilitation healthcare professionals’ 53 higher education (encompassing undergraduate and postgraduate degree courses for healthcare 54 professionals in rehabilitation) is challenging.”

Authors’ Comment:

We have revised the sentence for improved clarity.

Authors’ Action:

Amended as suggested. 

---

Reviewer #3 Comment:

“2. Line 72:

• Delete ‘starting’ as this is the only perspective explored in this article, correct?”

Authors’ Comment:

Amended as suggested. 

---

Reviewer #3 Comment:

“3. Line 91:

• Provide definition (length of time) for ‘recent’ graduate or include in demographic information in results section?”

Authors’ Comment:

Thanks for this comment. We have now reported that they were all graduated within a year or still students. 

Authors’ Action:

Page 6, lines 90-4 have been changed:

“To be included in the study, participants had to be health professionals working in rehabilitation and either current students or recent graduates (within the past year) of the MSc in 'Rehabilitation Sciences of the Health Professions' at the University of XXX.”

Reviewer #3 Comment:

“4. Lines 112- 114:

• At the beginning of this paragraph you only mention three researchers: SB, LF, student. Then in these lines you state:

“This folder was accessible from all researchers but LF and SB, so they would not know the names of the students who decided to take part in the study and the content of their interviews until they had been transcribed and anonymised.”

Are there other researchers or should this state: This folder was accessible only to the student researcher, to provide anonymity from LF and SB who are members of faculty.”

Authors’ Comment:

We would like to thank the reviewer for helping us to clarify this concept. We have further clarified it by adjusting the sentence according to the provided suggestions.

Authors’ Action:

The following sentence in the ‘Data Collection’ section, page 7, lines 111-5, has been adjusted as follows: “This folder was accessible only to the MSc student (NM) until the interview transcripts were transcribed and anonymised as LF and SB are faculty members. Once this process was over, access to this folder was granted to the other research team members (LF, BG, MT, AD, GB and SB).”

Reviewer #3 Comment:

“6. Line 129:

• This is the first mention of ‘BG’. Please provider earlier – I assume this is the initials of the student? Also remove ‘a’ before ‘BG’.”

Authors’ Comment:

We appreciate that the reviewer brought this unclear information to our attention. Benedetto Giardulli (BG) is a PhD student in neurosciences who analysed the dataset, but he did not conduct the interviews, which is why he was not mentioned earlier. Interviews were carried out by a male student (NM) who performed only this task.

To make it clearer, as for the other researchers like SB and LF, we have specified that Benedetto Giardulli (BG) is a PhD student in neurosciences and that the male student Nicolo Magistrelli conducted the interviews.

Authors’ Action:

The sentences in the ‘Data Analysis’ section, page 7-9, has been changed as follows: 

• Line 113: “[…] by a male student of the abovementioned MSc (NM)”

• Lines 138: “The themes were generated by BG, a PhD student in Neurosciences, with SB revising the whole process and reflecting upon the themes.”

---

Reviewer #3 Comment:

“7. Lines: 125- 129 and 132- 137:

• I feel the methodology could be more clearly articulated in these lines.

• I commend your commitment to reflexivity and keeping a reliable audit trail.

• Table 2 is also a clear and helpful way to document your qualitative methods and the contribution of each author.”

Authors’ Comment:

We would like to thank the reviewer for supporting our choice to conduct our analysis following the steps of the Reflexive Thematic Analysis by Braun and Clarke. We have addressed what has been suggested by the reviewer by improving the clearness of certain sentences of the ‘data analysis’ section.

Authors’ Action:

The following paragraphs in the ‘Data Analysis’ section, pages 8-9, lines 124-31 and 136-42 have been adjusted as follows:

• “The exploratory theories informing the analysis were an ‘experiential qualitative’ within a ‘realist theoretical’ framework as we intended to explore and understand the expected characteristics of the CL to reflect the perception of social reality (rehabilitation healthcare professionals), and to take the reality as voiced in the dataset. In this sense, themes are not in the data waiting, they are developed by exploring the intersection of the data and the researchers’ positioning, skill and interpretative work”;

• “We used an inductive approa

---

## [Editor Report · Decision Letter 1]

16 Oct 2024

Expected Features of the Course Leader in the Rehabilitation Healthcare Professionals’ Higher Education: A Qualitative Study on Students’ Perspectives

PONE-D-24-09846R1

Dear Dr. Battista,

We’re pleased to inform you that your manuscript has been judged scientifically suitable for publication and will be formally accepted for publication once it meets all outstanding technical requirements.

Kind regards,

Weifeng Han, PhD

Academic Editor

PLOS ONE

---

## [Editor Report · Acceptance letter]

21 Nov 2024

PONE-D-24-09846R1 

PLOS ONE

Dear Dr. Battista, 

I'm pleased to inform you that your manuscript has been deemed suitable for publication in PLOS ONE. Congratulations! Your manuscript is now being handed over to our production team.

Kind regards, 

on behalf of

Dr. Weifeng Han 

Academic Editor

PLOS ONE